# Moderating Effect of Gender and Engineering Identity on the Association between Interpersonal Relationships and Mental Health of Female Engineering Students

**DOI:** 10.3390/ijerph191610425

**Published:** 2022-08-21

**Authors:** Liang Wang, Xiangyu Zhou, Wei Wu, Aihua Chen

**Affiliations:** 1School of Public Affairs, Zhejiang University, Hangzhou 310058, China; 2School of Law and Politics, Zhejiang Sci-Tech University, Hangzhou 310008, China

**Keywords:** interpersonal relationship, mental health, female engineering students, gender identity, engineering identity

## Abstract

Influenced by factors such as gendered masculine culture within the engineering fields, female engineering students are facing increasing mental health issues. However, the effect of gender or engineering identity on the mental health distress of female engineering students was not well explored till now. This study adds to the current body of knowledge of mental health distress in female engineering students by proposing and verifying a moderating model based on social identity theory (SIT). The data were collected in June 2022 using a cross-sectional survey questionnaire distributed at five universities in eastern China (*N* = 376). A stepwise multiple regression analysis was performed to understand the relation between the tension of interpersonal relationships, the mental health distress female engineering students suffer from, and their gender or engineering identity. In our sample, 13.03%, 15.96%, and 14.36% of the female engineering students self-reported moderate to extremely severe stress, anxiety, and depression, respectively. Meanwhile, our results provide empirical evidence for the significantly positive relationship between the female engineering students’ tension of interpersonal relationships and their mental health distress, including stress, anxiety, and depression. Moreover, we found that gender identity can enhance the positive relationships mentioned above, while engineering identity could weaken these relationships. These findings provide empirical evidence for the role of social identity theory in dealing with mental health problems among engineering students. Broadly, the results of this work inform that social identity and professional role identity should be considered when designing interventions to prevent mental health crises among college students.

## 1. Introduction

Mental health is a non-negligible part of our health, just as was clarified by the World Health Organization in their summary report “There is no health without mental health” [1], which became the focus of attention regarding public health [2]. In recent years, mental health issues among college students (the average age of college students in this research is 21.86 years: minimum age = 19, maximum age = 28, SD = 1.65) received significant attention [3,4]. Poor mental health is usually associated with some negative consequences, such as a lower level of education, poor academic performance, self-harm, or even suicide [5]. For example, China’s national mental health development report (2019–2020) showed that 18.5% of college students had a tendency toward depression, among which 4.2% had a high risk of depression and 8.4% had a tendency toward anxiety. Meanwhile, female college students have a higher level of mental health problems, which means there is a significant gender difference in the mental health status of college students [6]. Influenced by complex factors, such as a masculine culture and disciplinary difficulty, mental health is becoming an issue of increasing concern in the field of engineering across the world. Results from a recent study found that stress and specific mental wellness issues are particularly acute in the field of engineering and twice the percent of engineering students were experiencing anxiety disorders (part of mental health symptoms) compared with other college populations [7]. In addition, in a survey conducted by Equal Engineers with 875 engineers in the UK, over a third of the engineers (37.7%) reported fair or poor mental health conditions, and a fifth of the engineers (22.0%) had to take time off from work because of their bad mental health [8]. These figures indicate an emergent need to care about mental health in the engineering profession.

Engineering is often aligned with a male-dominated and gendered masculine culture [9], and women or female students in engineering fields were long underrepresented and marginalized [10]. Recent statistics show that just 16.5% of people working in the engineering field are women in the UK. The ASEE number report shows similar figures for the USA (2012–2020) (Figure 1), displaying a rising trend but still a lower percentage of female engineering students at all three academic engineering and engineering technology degree levels [11]. In terms of gender study for women in engineering professions, a previous study indicated that the most important aim of relevant research was to achieve a sense of belonging to engineering, named *engineering identity* in the literature [12]. Recently, engineering identity was used in a growing body of research as a lens to investigate and explain the phenomena and impact of the underrepresentation of women groups in the gender-typed engineering fields, including the mental health distress faced by women engineering students. For example, Jensen and Cross’s research work investigated the relationships among mental health, engineering identity, and the sense of inclusion based on engineering stress culture, and their research demonstrated that engineering identity is related to students’ mental health [13]. On the basis of previous studies, this study aims to enhance the understanding of the relationships among gender identity, engineering identity, and mental health conditions in female engineering students.

### 1.1. Literature Review 

Over the past few decades, college students began facing a crisis of mental health, with a rising trend in the prevalence and severity of mental health problems [14], and concerns about the mental health crisis that engineering students face also grew in recent years [15]. High levels of stress and anxiety are common among engineering students, for example, a study of 1203 undergraduate students enrolled in the college of engineering at a large, public university found that 12.8%, 9.9%, and 14.1% of the participants reported depression, stress, as well as anxiety levels categorized as severe or extremely severe, respectively [16]. Moreover, in a survey by Danowitz and Beddoes, 28.4% of engineering students from five institutions across the western U.S. reported potential diagnosable mental health problems and 55.2% of the students screened positive for moderate psychological distress [17].

Some studies examined the negative impact of mental health on students’ retention and success [18]. A study conducted by Kruisselbrink identified six factors impacting students’ mental health: academic pressure, financial burden, increased accessibility to higher education, increased female-to-male student ratio, increased use of technology, and a dramatic change in the lifestyle of university and college students [19]. Marano pointed out that broken families and increased competitiveness are the main reasons for the mental health issues college students face [20]. Other research also revealed that students’ mental health can be affected by many demographic and social factors [14]. Among these vital factors, students’ interpersonal relationships attracted many researchers’ attention. For example, Stewart-Brown reviewed the significance of interpersonal relationships in influencing mental health [21]. Okada et al. identified a positive relationship between interpersonal relationships and depression or less satisfaction with school life among students [22]. Li et al. found that adolescents’ mental health problems are highly associated with interpersonal relationships in school [23]. However, the effect of interpersonal relationships on mental health in engineering fields is not fully understood.

Furthermore, plenty of research reported that female students are more likely to suffer from mental health disorders compared to the male population. Based on a survey of 700 first- and second-year engineering students at five universities across the western United States, Danowitz and Beddoes found 28.4% of respondents potentially suffered from a diagnosable mental health condition, and non-white female engineering students showed higher rates of panic and PTSD disorders compared with their male peers [17]. A study conducted by Foster and Spencer also found that female engineering students were significantly more likely than males to have a higher level of stress, with correlation and chi-square analyses (r = 0.229 and *p* = 0.019) [24]. Levels of stress and anxiety are also significantly higher for female than male engineering students in a recent study conducted at a large public research-intensive institution in the Midwest United States [13]. Moreover, a previous study found engineering students with mental health disorders were usually less likely to receive treatment compared with students from other academic disciplines [25]. Overall, the results and findings from these studies indicate that female engineering students’ mental health status is an important research field needing careful attention.

Social identity, more specifically engineering identity, was seen as an alternative consideration in developing proactive interventions for female engineering students’ mental health distress [13]. According to Godwin’s research work, engineering identity presents the understanding of how engineering students view themselves, are viewed by others, and view their role in society, which is composed of performance/competence, recognition, and interest in an engineering-related domain [26]. In Jensen and Cross’s work in progress, engineering identity was found to be negatively correlated with the depression level among engineering students in the sample population [16]. Additionally, engineering identity was related to the perceptions of inclusion within academic departments (department caring, pride, and diversity), which was proven to have a relationship with the mental health of engineering students [13]. However, the specific effect of gender identity or engineering identity on the relationships mentioned above is still not well understood, especially for female students in engineering disciplines. To bridge the gap, this study will use an identity negotiation analytical lens to expand the knowledge of the effect of gender identity and engineering identity on the mental health (stress, anxiety, and depression) of college engineering students.

### 1.2. Theoretical Framing

In this paper, we use social identity theory to guide and interpret our research design and findings. Social identity theory (SIT) is a broad social psychology theory regarding the role of self and identity within a group and intergroup interactions [16], which was originally developed by Henri Tajfel [27]. Social identity theory is used as a theoretical perspective to better understand the engagement, persistence, and retention of students in science, technology, engineering, and mathematics (STEM) fields [28]. In addition to social identity theory, some researchers explored the interrelationships among multiple identities with the construct of social identity complexity (SIC) [29] and the negotiating process among these various sociocultural group-based or unique personal-based identities (identity negotiation theory, INT) [12,30]. Engineering identity, a specific type of social identity in engineering education research fields, was defined as the way in which students describe themselves and are positioned by others in the role of being an engineer [31]. Based on SIT and the research work conducted by Ross et al. [28], we divided the identity into social identity (referring to the constructs of race/ethnicity or gender, namely gender identity in this study) and role identity (referring to the roles that a person played in their work environment, namely engineering identity in this study). Female engineering students are often at the intersection of social identification, such as gender identity, and professional role identification, such as engineering identity.

Most female engineering students face conflicts between gender identity, which can be viewed as a gender stereotype (for example, “I am not appropriate for engineering work because of my gender”) [32], and engineering identity, which can be seen as a sense of belonging to professional culture (for example, “I define myself through a role or performance in engineering”) [33]. Hatmaker found that gender stereotypes in engineering environments usually place professional identity on the periphery, while overly validating gender identity [12], which means female engineers could be viewed as women first and engineers second [34]. In such situations, women engineers can use impression management and coping strategies to construct their professional identity [12], which means in some way that by constructing an engineering identity or building a sense of belonging to the engineering profession, gender and professional identity can be negotiated to deal with the conflicts among women engineers. Previous studies also demonstrated that the construction of engineering identity has important implications for career choices [35], professional attainment and persistence [36], learning motivation [37,38], and the engineering interest [31] of engineering students. However, little is known about the role of gender and professional engineering identity negotiation in mental health among women engineering students. The purpose of this study was to test and clarify the moderating effect of gender and engineering identity on the relationship between the tension of interpersonal relationships among, and mental health of, female engineer students in China, as well as to provide some data-based evidence for policy makers to improve the mental health conditions of engineering students. As the most commonly identified mental health disorders among college students were depression, anxiety, and stress-related concerns [18], we mainly measured mental health by these three dimensions.

### 1.3. Research Hypotheses and the Conceptual Model

On the basis of the research questions and current academic theories, we put forward a research model and the following hypothetical relationships among different variables in this study:

**Hypothesis** **1:***Female engineering students’ tension in interpersonal relationships is positively related to the levels of their mental health distress, measured by stress (H1a), anxiety (H1b), and depression (H1c), respectively*.

**Hypothesis** **2:***Female engineering students’ gender identity positively moderates the relationship between the tension of interpersonal relationships and the level of mental health distress (H2a, H2b, and H2c), which means that the female engineering students’ level of stress, anxiety, and depression will be more easily affected by their poor interpersonal relationships when they attain a higher level of gender identity*.

**Hypothesis** **3:***Female engineering students’ engineering identity negatively moderates the relationship between the tension of interpersonal relationships and levels of mental health distress (H3a, H3b, and H3c), such that when engineering identity is stronger, female engineering students’ tension in interpersonal relationships will have a weaker effect on their risk of mental health distress*.

On the basis of these hypothetical relationships, we drew the conceptual model of our research (Figure 2).

## 2. Materials and Methods

### 2.1. Measures

Four measuring instruments were used in our questionnaire to measure each of these constructs in addition to demographic information (Appendix A Table A1).

In research conducted by Jensen and Cross (2020), mental health distress of female engineering students was measured by the Depression Anxiety Stress Scales 21 (DASS-21), with 7 items for each stress, anxiety, and depression subscale [39]. DASS-21 is a well-established instrument for measuring negative mental health status with good reliability and validity [40]. Responses for DASS-21 were created on a 4-point Likert scale that ranged from 0 (*It does not fit my situation at all*) to 3 (*It completely fits my actual situation*). We multiplied the original scores obtained from questionnaires for each subscale by 2 in order to calculate the final scores in Table 1 by comparing them with the cutoff values recommended on the DASS official web page (http://www2.psy.unsw.edu.au/groups/dass/, accessed on 22 May 2022). In our research, a score of 1 to 5 (displayed in Table 1) means normal to extremely severe mental health conditions. The lower scores indicate a lower level of stress, anxiety, or depression, indicating a better status of mental health among female engineering students.

Referring to the study conducted by Li et al. [23] and Darling et al. [41], as well as our interview with some female engineering students, we developed a new instrument to measure the degree of tension regarding female engineering students’ interpersonal relationships. The instrument is comprised of 4 questions, and the participants were asked to evaluate their relationships with their teachers, classmates, parents, and other friends by answering questions, such as “*What do you think of your relationship with your teachers or supervisors?*” Each item of the instrument was rated by the student on a five-point scale, ranging from 1 (*Very harmonious*) to 5 (*Very tense*). Therefore, in our instrument, a higher score meant that the female engineering student was suffering from more tense interpersonal relationships.

For moderating variables, gender identity was measured using a scale adapted from the Sex Role Attitudinal Inventory (SRAI) [42,43]. The subscale included 5 items, for example, “*Most men are better suited for engineering than most women.*” Each item was rated on a Likert scale from 1 (*Strongly disagree*) to 5 (*Strongly agree*), with higher scores indicating a stronger sense of gender stereotype in engineering professions. Meanwhile, engineering identity was measured using a 10-item scale initially developed by Allison Godwin [44], containing three different sub-dimensions, interest, recognition, and performance/competence, which were validated in research conducted by Godwin and her colleagues [31]. In our study, we did not distinguish among the three sub-dimensions, but used a comprehensive construct to capture the overall state of engineering identity. Each item in the engineering identity scale was shown in a 5-point Likert format ranging from 1 (*Strongly disagree*) to 5 (*Strongly agree*), a higher score meaning a sense of higher belonging to or self-identification with engineering professions.

Covariates in this study contained school type (ST, generally, is based on the MOE’s classification; there are three types of university in China: 985 institutions are at the top level, 211 institutions are at the secondary level, and others are ordinary institutions), parents’ education levels (PE), family economic status (ES), and academic workload (AW). 

To confirm the internal consistency, we calculated Cronbach’s alpha coefficients for each construct, where higher values indicate increased reliability of the subscale [45]. The DASS-21 and engineering identity’s Cronbach’s alpha coefficients were above 0.9, while we originally developed TIR measurement instrument and gender identity scale adapted from SRAI were 0.82 and 0.83, respectively. Overall, the reliability values in Table 2 show that each subscale had internal consistency scores above 0.80, which fall within the acceptable range of reliability standards in educational research [46].

### 2.2. Participants

The data used in this study were gathered in June 2022 from a cross-sectional online survey conducted among engineering students. We applied a cluster sampling plan and e-mailed survey invitations with a questionnaire link to more than 2500 engineering students from 5 different types of universities in eastern China. In all, 500 questionnaires were distributed to each university, including two comprehensive universities (both are 985 institutions) and three polytechnic universities (including one 211 institution and two ordinary institutions). We also posted our survey link on a within-campus forum at one 985 institution in order to recruit more participants for our research. In addition, in the e-mail, as well as the preface of the questionnaires, we clarified the purpose of this survey and promised, in bold font, to keep the information of all participants confidential. We offered a modest financial incentive (about 5 *yuan* RMB) for the one who finished the questionnaire. A total of 1,152 questionnaires were answered (among them, about 150 were from the within-campus forum), so the response rate was approximately 40% in our study. The percentage of female engineering students participating in our survey was about 36% (415 responses from female engineering students and 737 from male engineering students or those who did not report their gender). Some responses were removed on the basis of the following criteria: those who did not complete 100% of the questionnaire, those whose response time was extremely short (less than 3 min), or those whose answers to some similar items were obviously inconsistent. Overall, the responses of 376 female participants were included in our following analysis.

### 2.3. Statistical Analyses

All statistical analyses were performed with Stata14MP (StataCorp, College Station, TX USA). The statistical analysis process was as follows: (1) Descriptive analyses were conducted for every construct, and the percentages of self-reported stress, anxiety, and depression by the level of severity were also calculated. (2) Regression analyses were performed with the ordinary least square method to identify the relationship between the quality of female engineering students’ interpersonal relationships and their mental health distress conditions. (3) A stepwise multiple regression analysis was performed with interaction terms TIR*GI or TIR*EI to clarify the moderating role of gender identity and engineering identity in the correlation between the tension of interpersonal relationships and mental health distress of female engineering students. In addition, we further visualized the moderating effects of gender and engineering identity with the help of the marginsplot syntax command within Stata.

## 3. Results

### 3.1. Descriptive Statistics

We calculated the population and percentage of self-reported stress, anxiety, and depression by the level of severity (Table 3). In our sample, most respondents reported a normal level of mental health distress; 13.03% of the female engineering students reported moderate to extremely severe stress, 15.96% reported moderate to extremely severe anxiety, and 14.36% reported moderate to extremely severe depression. The percentages reported by our samples are lower than the levels of self-reported stress, anxiety, and depression in previous studies (in Jensen and Cross’s study, the percentages were 28.97%, 35.81%, and 34.92%, respectively, for each mental health distress dimension). Meanwhile, we compared the female engineering students’ mean score of mental health with the male or not-reported gender participants’ results of the same questionnaire. Similar to previous research [13,17,24], our results show that female engineering students tend to have a higher mean score of mental health disorders than men or not-reported gender engineering students based on the DASS-21 (female: 1.46, 1.61, and 1.56; male or not-report gender: 1.43, 1.55, and 1.51; for mean score of stress, anxiety, and depression, respectively). Furthermore, chi-square tests of independence were performed to examine the relationship between gender and the severity levels of stress, anxiety, and depression. We found that there were no significant difference between genders in terms of anxiety and depression severity level, but there was a significant relationship between gender and stress level (*X*^2^(4) = 4.121, *p* = 0.039, and *N* = 1,113, among which 376 were female and 737 were male or not-reported)—the same findings as Foster and Spencer’s findings [24].

The descriptive statistic results of ONLY female participants are presented in Table 4. As far as school type goes, more than 20% of the respondents were from 985 or 211 institutions (16.22% and 7.98%, respectively), while 73.4% of the respondents were from ordinary institutions. Although the average value of all samples showed that students’ mental health distress scores were between normal and mild levels, it is obvious that respondents from 211 institutions had slightly higher average mental health distress scores (1.73, 1.93, and 1.90 for stress, anxiety, and depression, respectively). The education level of the parents of more than 85% of the respondents was above high school, and the respondents of parents with a high school-level education had the highest score in mental health distress, tension of interpersonal relationships, and gender identity, while the same respondents had the minimum score for engineering identity. More than 80% of the respondents were self-reportedly from families with a middle to superior economic status, with the middle level taking up the largest proportion when comparing other economic conditions. Furthermore, there is deemed to be a negative correlation between economic status and mental health state, as the female engineering students in our sample from a superior economic status tended to have mild to moderate mental health distress scores (2.02, 2.12, and 2.10 for stress, anxiety, and depression, respectively). Meanwhile, the female engineering students from a superior economic status had the highest level of tension in terms of interpersonal relationships and gender identity, as well as the lowest level of engineering identity. As for academic workload, which was viewed to be the best predictor of students’ mental health problems [47], nearly 80% of the female engineering students in our sample reported hardly any or little. Moreover, in our study, the academic workload may not be a strong explanatory variable for engineering students’ mental health distress because the respondents who reported quite a lot of academic workload had a lower score for stress, anxiety, and depression than the three-dimensional average value. Overall, the mental health distress of our total sample was between normal and mild levels (1.46, 1.61, and 1.56 for stress, anxiety, and depression, respectively), with scores of 1.97, 2.04, and 3.89 for tension of interpersonal relationships, gender, and engineering identity, which were measured on 5-point Likert scales.

### 3.2. Main Effects

Regression analyses were conducted to test the main effect of the tension of interpersonal relationships on the mental health distress of female engineering students, such as stress, anxiety, and depression (Table 5). The adjusted R-squared in each model exceeded 0.6, indicating that the explanation of the effect of independent variables in our models was accepted [44]. Significance coefficient values with *p* < 0.001 in models (1), (3), and (5) suggested that there was a strong main effect, as we assumed in hypothesis 1. For a more robust analysis, we added some control variables, such as academic workload, school type, parents’ education levels, and economic status, to the regression analyses in models (2), (4), and (6). Slightly lower, but still significant, coefficient values indicated the strong correlations between the independent variable and dependent variables. On the basis of these results, we found that the female engineering students in our sample who have poor interpersonal relationships with their teachers, parents, schoolmates, and friends tend to suffer from higher levels of mental health distress. In addition, for the different symptoms of mental health distress, such as stress, anxiety, and depression in the current study; higher coefficient values meant that female engineering students with more tense interpersonal relationships have a higher level of depression than stress or anxiety. Therefore, we can conclude that hypothesis 1 is supported.

### 3.3. Moderating Effects

At this stage, we concentrated on the coefficients of interaction in terms of the tension of interpersonal relationships and gender identity (TIR*GI) or engineering identity (TIR*EI). If the coefficients of interaction terms were significant, we could infer that moderating effects existed. Moreover, in this research, we regarded the Likert scale as a continuous variable, so we calculated interaction terms by a widely used procedure proposed by Aiken and West [45]. Meanwhile, considering that the delta R squares are too small in regression equations with and without control variables (Table 5), we ignored the control variables in our moderating effect regression models. Table 6 presents the results of moderating regression analysis results. The coefficient values of the interaction terms TIR*GI were all positive and significant, at the level of *p* < 0.001, in models (7), (9), and (11), indicating that there were positive moderating effects of gender identity on the relationships between TIR and stress, anxiety, and depression. These results show that the level of stress, anxiety, and depression among female engineering students will be more easily affected by their poor interpersonal relationships when they attain a level of gender identity, meaning hypothesis 2, proposed earlier, is supported by our data. Meanwhile, the coefficient values of the interaction terms TIR*EI were also significant, at the level of *p* < 0.001, in models (8), (10), and (12), but the coefficients were negative values, suggesting that the poor interpersonal relationships of female engineering students have a weaker effect on their risk of mental health problems when their engineering identity is strong. Overall, hypothesis 3 was supported as well.

Figure 3 visualizes the moderating effect of gender identity more clearly. The regression lines for a high gender identity and a low gender identity within the relations between the tension of interpersonal relationships and stress, anxiety, and depression among female engineering students are presented, depicting that the independent variable (TIR) and the dependent variables (stress, anxiety, and depression) are positively correlated. In addition, the lines of high gender identity are steeper than those of low gender identity, pointing to a clear positive moderating or promoting effect of gender identity on the tension of interpersonal relationships and the mental health distress relations of female engineering students at 5 universities in eastern China.

The same regression lines for high engineering identity and low engineering identity are also presented in Figure 3. Overall, there are different linear trends for female engineering students with high engineering identity and low engineering identity. Specifically, the line of low engineering identity indicates a positive correlation between the tension of interpersonal relationships and mental health distress, while the line of high engineering identity means a negative correlation between the independent variable and dependent variables. In other words, engineering identity has a negatively moderating effect on the association between the female engineering students’ tension of interpersonal relationships and their levels of stress, anxiety, and depression, as we stated in hypothesis 3 (H3a, H3b, and H3c) and showed in the conceptual model (Figure 2).

## 4. Discussion

During the past few decades, the mental health of college students attracted the attention of many researchers, and there is more of growing concern about the mental health problems of female students than those of their male peers [3,4,6]. Meanwhile, engineering students are seen as more vulnerable to mental health problems than students in other fields because of the male-dominated and gendered masculine culture [7]. Therefore, the mental health conditions of female engineering students were noticed in engineering education research [13]. However, while most studies concentrated on the reasons and impacts of mental health distress in female engineering students, the role of the identity of female students was not previously explored in detail. By performing this social identity theory-based study, we found: (1) The levels of self-reported mental health severity for female engineering students in our study were lower than Jensen and Cross’s findings based on American students: 13.03%, 15.96%, and 14.36% of the sample students in our research reported moderate to extremely severe stress, anxiety, and depression separately, and 28.97%, 35.81%, and 34.92% for the same dimensions in Jensen and Cross’s work [13]. Furthermore, the chi-square test for independence showed there were significant differences between gender and stress levels (*X*^2^(4) = 4.121 and *p* = 0.039) while there were no significant differences between gender and anxiety or depression levels (*X*^2^(4) = 2.056, *p* = 0.725; *X*^2^(4) = 2.413, *p* = 0.660), which were slightly different with Jensen and Cross’s finding, in which gender was significantly related to self-reported stress and anxiety levels, but not significantly related to self-reported depression levels (*X*^2^(4) = 27.92, *p* < 0.001 *; *X*^2^(4) = 21.169, *p* < 0.001 *; and *X*^2^(4) = 5.522, *p* < 0.238 *). (2) This research work confirmed the correlation between the tension of interpersonal relationships and mental health distress conditions among college students in engineering professions, which were findings in some ways consistent with previous research [21,22,48]. However, the regression coefficients of the main effect models without controls were 1.087, 1.146, and 1.185 for the tension of interpersonal relationships on stress, anxiety, and depression, respectively, which were higher than Darling et al.’s results (−0.161, −0.082, −0.070, and −0.132 for friendship, parental relationship, love relationship, and family relationship on sense of coherence, respectively [41]). These comparisons illustrated that interpersonal relationships may play a more important role in mental health for Chinese college students than American college students. Nevertheless, the coefficients of the tension of interpersonal relationships on stress in our study were lower than the regression coefficients of academic workload (0.211), separation from school (0.324), as well as fears of contagion (0.121) on perceived stress in Yang et al. research during the COVID-19 pandemic [47]. (3) In this study, we further proposed a moderating model with different identities to better understand the role of identity in mental health distress (stress, anxiety, and depression) in female engineering students, and clarified the positive and negative moderating roles of gender and engineering identity, respectively. Specifically, we found that the mental health distress of female engineering students with a higher gender identity were more easily affected by their poor interpersonal relationships (positive moderating effect were 0.225, 0.303, and 0.320 for stress, anxiety, and depression, respectively), while a higher engineering identity can weaken these correlations (negative moderating effect were −0.388, −0.493, and −0.402 for stress, anxiety, and depression, respectively), which may contribute to the knowledge of gender research and professional identity research.

The findings of this work have some implications for policymakers, faculty members, or college administrators in engineering education fields. Collectively, given the underrepresented and marginalized image of females in the male-dominated and gendered masculine culture of engineering fields [9], more attention should be focused on the mental health distress of female engineering students. Meanwhile, both gender identity and professional engineering role identity should be considered when designing interventions to prevent mental health crises among female engineering students. For example, constant recognition from teachers or parents, which is usually seen as a sub-dimension of engineering identity [31,35,44,49], may ease the harm of mental health distress.

The generalizability and application of our work are limited in some ways: (1) The sample we used in our study was small (*n* = 376) and the participants were from five universities in eastern China, which restricted the generalizability and applicability of our findings in other contexts. (2) We only verified the positive correlations between the tension of interpersonal relationships and mental health distress among female engineering students with the OLS regression methods and our cross-sectional data. However, any causality between these variables cannot be proved in this work. Female engineering students may have mental health illnesses because of poor interpersonal relationships, while the mental health distress may also influence the development of their social skills and ability to maintain good relationships [50]. (3) Although we presumed and investigated the moderating effect of gender identity and engineering identity on the relationship between the tension of interpersonal relationship and mental health distress in female engineering students, we did not discuss how these effects occur and how to negotiate these different identities in order to better deal with mental health a crisis.

For future directions, on the one hand, we need to expand the scale and representativeness of our research samples and further examine the specific mechanisms behind the roles of identities within different types or multiple professional fields in future research work. On the other hand, accurate causality among different variables should be investigated through more robust methods and serial or panel data. In addition to the socioeconomic factors, some mind related factors, such as the engineering students’ degree of neurodiversity, could also be considered in future research. Furthermore, we need to extend this study into a qualitative research design of particular female engineering students’ experiences with the effect of their gender or engineering professional identity on their mental health status. Through this qualitative study, we can deeply observe the whole process and indicate which students in the engineering field may be specifically ’at risk’ in terms of mental health. Meanwhile, proper qualitative study, based on a specific person or accident, would help us enhance our understanding of why these effects showed in Figure 3. occur, and how to negotiate these different identities in order to better deal with the mental health crisis. As a result, some more targeted and proactive interventions for the mental health distress of female engineering students, based on the roles of gender and engineering identity, could be developed or discussed in future research.

## 5. Conclusions

Although more and more research demonstrates that engineering students may be more vulnerable than others to mental health and wellness issues [17], the mental health of female students in male-dominated or gendered masculine fields, such as engineering, is still often understudied, even though many studies focused on the mental health distress of college students or engineering students. The correlations between engineering students’ social or role identities and their mental health conditions were well explored, either.

On the basis of the theory of social identity, in this study, we identified social identity (referring to the constructs of race/ethnicity or gender) and role identity (referring to the roles that a person plays in their work) to describe the intersection of social identification and professional role identification, respectively, of female engineering students. Using the data collected from a cross-sectional survey at five universities in eastern China, we provided quantitative empirical evidence for the positive relationships between the tension of interpersonal relationships and the levels of stress, anxiety, and depression in female engineering students. We found that gender identity can enhance the relationship mentioned above, while engineering identity can weaken these correlations. The proposed moderating model could provide empirical support for, as well as enhance, the current understanding of the effect of the identity of engineering students on their mental health status. Moreover, the results of this study could contribute to the knowledge of engineering education research, as well as identity research. Further, proactive interventions for mental health disorders of female engineering students could also be designed and developed based on our findings.

## Figures and Tables

**Figure 1 ijerph-19-10425-f001:**
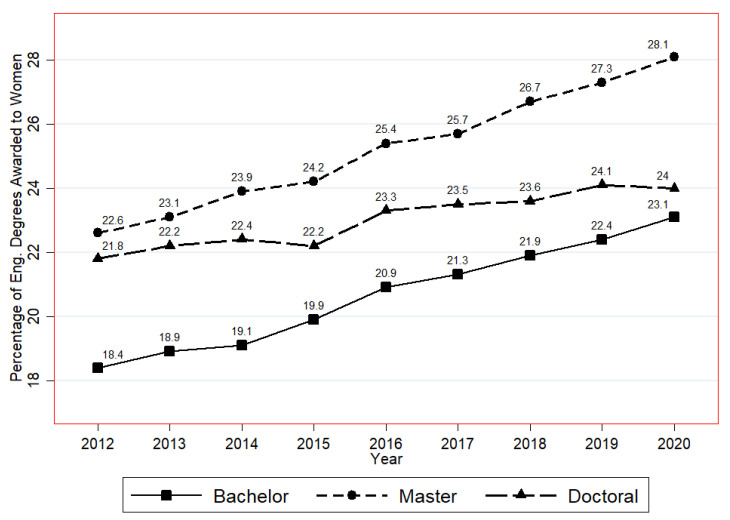
Percentage of bachelor, master, and doctoral degrees awarded to women in engineering and engineering technology in the USA (2012–2020) [11].

**Figure 2 ijerph-19-10425-f002:**
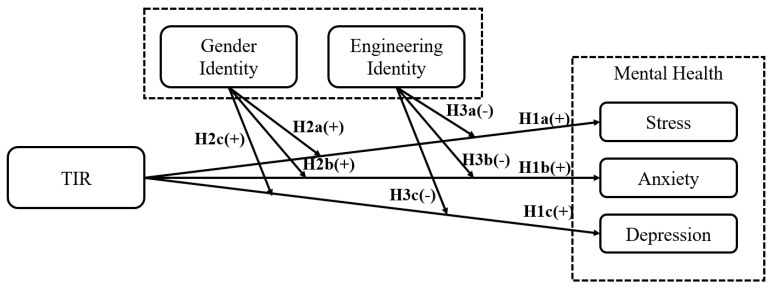
Conceptual model of possible relationships among the tension of interpersonal relationships, mental health distress, gender identity, and engineering identity.

**Figure 3 ijerph-19-10425-f003:**
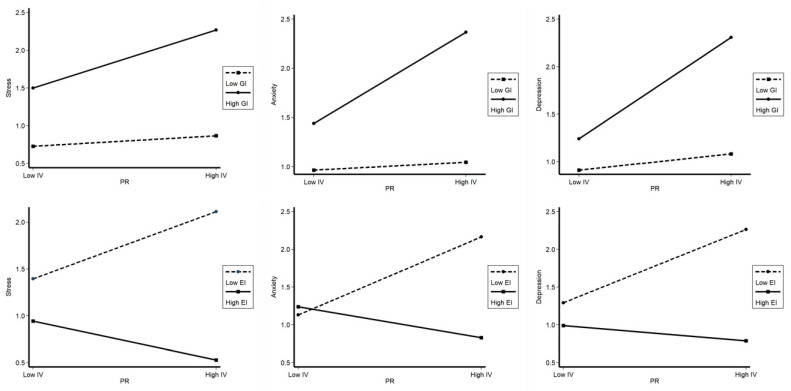
Visualizing the moderating effect of gender identity (GI) and engineering identity (EI).

**Table 1 ijerph-19-10425-t001:** Score and cutoff values on the DASS-21.

Meaning	Stress	Anxiety	Depression	Score
Normal	0–14	0–7	0–9	1
Mild	15–18	8–9	10–13	2
Moderate	19–25	10–14	14–20	3
Severe	26–33	15–19	21–27	4
Extremely severe	34+	20+	28+	5

Note: The score is equal to twice the original value obtained from the questionnaire.

**Table 2 ijerph-19-10425-t002:** Score and cutoff values on the DASS-21.

	Constructs	Scale	Cronbach’s Alpha
Independent variable	TIR	Originally developed	0.820
Dependent variables	Stress	DASS-21	0.925
	Anxiety		0.917
	Depression		0.911
Moderating variables	Gender identity	Adapted from SRAI	0.833
	Engineering identity	Engineering identity scale	0.941

Note: TIR = tension of interpersonal relationship; Cronbach’s alpha for completed items in DASS-21 is 0.968.

**Table 3 ijerph-19-10425-t003:** Self-reported stress, anxiety and depression levels by gender groups, *n* (%).

	Gender	Normal	Mild	Moderate	Severe	Extremely Severe	Chi-Square Test for Independence
Stress	Female	318 (84.57%)	9 (2.39%)	10 (2.66%)	12 (3.19%)	27 (7.18%)	*X*^2^(4) = 4.121 *p* = 0.039 *
	Male or not-reported	632 (85.75%)	10 (1.36%)	30 (4.07%)	16 (2.17%)	49 (6.65%)	
Anxiety	Female	281 (74.73%)	35 (9.31%)	19 (5.05%)	6 (1.60%)	35 (9.31%)	*X*^2^(4) = 2.056 *p* = 0.725
	Male or not-reported	564 (76.53%)	75 (10.18%)	26 (3.53%)	10 (1.36%)	62 (8.41%)	
Depression	Female	292 (77.66%)	30 (7.98%)	17 (4.52%)	3 (0.80%)	34 (9.04%)	*X*^2^(4) = 2.413 *p* = 0.660
	Male or not-reported	575 (78.02%)	64 (8.68%)	37 (5.02%)	10 (1.36%)	51 (6.92%)	

Note: * *p* < 0.05.

**Table 4 ijerph-19-10425-t004:** Number and percentage of self-reported stress, anxiety, and depression by covariates (*N* = 376).

		*N*	%	Mental Health			TIR	GI	EI
				Stress	Anxiety	Depression			
ST	985	61	16.22	1.05	1.13	1.07	1.73	1.77	4.11
	211	30	7.98	1.73	1.93	1.9	2.21	2.26	3.83
	Ordinary	276	73.40	1.54	1.71	1.64	2.02	2.08	3.84
	Others	9	2.39	1.00	1.00	1.00	1.56	1.89	4.19
PE	Primary school	16	4.26	1.00	1.19	1.00	1.58	1.70	4.28
	Middle school	39	10.37	1.08	1.08	1.00	1.72	1.82	4.24
	High school	98	26.06	1.71	1.89	1.74	2.14	2.17	3.76
	Undergraduate	169	44.95	1.44	1.60	1.59	1.95	2.02	3.86
	Graduate	54	14.36	1.46	1.67	1.67	2.03	2.13	3.87
ES	Extremely difficult	15	3.99	1.00	1.13	1.00	1.65	1.89	4.27
	Difficult	56	14.89	1.25	1.41	1.32	1.83	1.86	4.06
	Middle	159	42.29	1.44	1.55	1.55	1.96	2.05	3.86
	Good	96	25.53	1.40	1.65	1.51	1.98	2.00	3.96
	Superior	50	13.30	2.02	2.12	2.10	2.26	2.30	3.54
AW	Hardly any	118	31.38	1.63	1.79	1.69	2.11	2.16	3.81
	Little	182	48.40	1.52	1.71	1.70	2.00	2.06	3.80
	Appropriate	29	7.71	1.03	1.14	1.00	1.76	1.77	4.24
	A lot	28	7.45	1.00	1.07	1.00	1.59	1.86	4.15
	Quite a lot	19	5.05	1.16	1.16	1.00	1.68	1.81	4.34
	Total	376	100	1.46	1.61	1.56	1.97	2.04	3.89

Note: TIR = tension of interpersonal relationship, AW = academic workload, ST = school type, PE = parents’ education level, ES = economic status, GI = gender identity, and EI = engineering identity; 985 and 211 institutions are seen as top and superior universities in China.

**Table 5 ijerph-19-10425-t005:** Results of regression analyses for the main effect.

	Stress		Anxiety		Depression	
	(1)	(2)	(3)	(4)	(5)	(6)
TIR	1.087 *** (0.044)	1.072 *** (0.045)	1.146 *** (0.048)	1.127 *** (0.049)	1.185 *** (0.042)	1.168 *** (0.042)
Controlling	No	Yes	No	Yes	No	Yes
Constant	−0.684 *** (0.094)	−0.990 *** (0.312)	−0.648 *** (0.103)	−0.941 *** (0.341)	−0.783 *** (0.090)	−1.281 (0.297)
F	615.46 ***	125.16 ***	572.48 ***	116.48 ***	801.66 ***	164.07 ***
Adj. R square	0.621	0.623	0.604	0.606	0.681	0.685
Observations	376	376	376	376	376	376

Note: *** *p* < 0.001; TIR = tension of interpersonal relationship; control variables: academic workload, school type, parents’ education levels, and economic status.

**Table 6 ijerph-19-10425-t006:** Results of regression analyses for moderating effect.

	Stress		Anxiety		Depression	
	(7)	(8)	(9)	(10)	(11)	(12)
TIR	−0.190 (0.133)	1.600 *** (0.112)	−0.319 * (0.155)	2.105 *** (0.137)	−0.286 * (0.129)	1.791 *** (0.108)
GI	0.211 (0.114)		−0.056 (0.133)		−0.163 (0.111)	
TIR*GI	0.225 *** (0.040)		0.302 *** (0.046)		0.320 *** (0.038)	
EI		0.177 (0.098)		0.618 *** (0.119)		0.278 ** (0.094)
TIR*EI		−0.388 *** (0.036)		−0.493 *** (0.044)		−0.402 *** (0.035)
Constant	0.381 (0.278)	0.381 (0.382)	0.979 ** (0.323)	−1.430 ** (0.467)	0.996 *** (0.270)	−0.197 (0.367)
F	422.27 ***	522.95 ***	337.72 ***	370.46 ***	503.19 ***	634.86 ***
Adj. R square	0.771	0.807	0.729	0.747	0.801	0.835
Observations	376	376	376	376	376	376

Note: * *p* < 0.05, ** *p* < 0.01, and *** *p* < 0.001; TIR = tension of interpersonal relationship, GI = gender identity, EI = engineering identity.

## Data Availability

Not applicable because of privacy issues.

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
