# Peer review of "Moderating Effect of Gender and Engineering Identity on the Association between Interpersonal Relationships and Mental Health of Female Engineering Students"

_ijerph, 2022, doi:10.3390/ijerph191610425_

Round 1

Reviewer 1 Report

The current study is clearly written and properly structured. All text is comprehensive and tables/figures are useful and readable. I believe that the paper add knowledge to the research field, is original, with recent references. The length of the paper is appropriate, although, it was possible, I would prefer that the discussion and conclusion sections were more extensive, because, in my opinion, the discussion section must be supported by own results using percentages.

Abstract:

When you say “this study fills the gap”, are you really sure about that?

Literature review:

Line 81 you could give more details about study of reference 15. The same for the ideas in lines 100 to 104, more details about these studies, to compare your own research with that in the discussion section.

Material and methods:

Line 194 “we developed a new instrument”. Are validated? How? Who? The questions of the new instrument are general enough and, in my opinion, it is not correct answer them using a scale, they are question that must be answered with a biographic text and analyse using categories.

Line 232 What are the different levels?

Line234 if you have the same questionnaire from male, why did you analyse it? In my opinion it is the weak point of your research. You must compare the female results with the male results of the same questionnaire. I think that your research could be improved if you make this male analysis.

Results:

Line 300, 301, 302 the results of 13.03, 15.96 and 14.36, how do you calculate? Why are they significant? In the table 4 are different values that I don’t understand.

Line 303, if your values are lower than other studies why are they extremely severe?

Line 362 Put together figures 3 and 4 for a better comparison and bigger.

Discussion and conclusions:

Discussion and conclusions are consistent with the actual results and the research question are answered, but a bit short (conclusions) and general (discussion and conclusions). I consider that you must expand them, because the comparison with previous works is not presented (value by value). It is very important compare studies value to value and not idea versus idea. The principal ideas must be showed in the conclusions, and the discussion section is to compare your values with values of previous works.

Line 419 what are the specific results of these studies?

Author Response

Dear editor and reviewer:

Thanks very much for reviewing this manuscript carefully. We really appreciate all your suggestions! Here are our itemized responses in below and my revisions in the resubmitted files.

Point 1:

I would prefer that the discussion and conclusion sections were more extensive, because, in my opinion, the discussion section must be supported by own results using percentages.

Discussion and conclusions are consistent with the actual results and the research question are answered, but a bit short (conclusions) and general (discussion and conclusions). I consider that you must expand them, because the comparison with previous works is not presented (value by value). It is very important compare studies value to value and not idea versus idea. The principal ideas must be showed in the conclusions, and the discussion section is to compare your values with values of previous works.

Response 1: Thank you for your comments. We have improved the discussion and conclusion parts according to your suggestions. On the one hand, we compared our results with other relevant findings value by value. However, we didn’t find other similar moderating models of gender or engineering identity, so we didn’t make such comparisons for the coefficients of moderating models. On the other hand, we expanded the conclusion part and made our key findings clearer.

Point 2: When you say “this study fills the gap”, are you really sure about that?

Response 2: Thanks! We find there are some research papers that have explored the mental health distress in female engineering students through an identity lens, for example, Jensen and Cross’s 2021 work. “Fills the gap” in the abstract is not appropriate, so we modified this sentence to “This study adds to the current body of knowledge on mental health distress in female engineering students by proposing and verifying a moderating model based on social identity theory (SIT)”.

Point 3: Line 81 you could give more details about study of reference 15. The same for the ideas in lines 100 to 104, more details about these studies, to compare your own research with that in the discussion section.

Response 3: For reference 15 (16 in the revised version), we add some more detailed information about their research findings with specific figures like “12.8%, 9.9%, and 14.1% of the participants reported depression, stress as well as anxiety levels categorized as severe or extremely severe, respectively” (Jensen and Cross, 2018). For lines 100 to 104, firstly, we add more detailed information about the studies mentioned in our literature review to demonstrate the necessity to care about female engineering students’ mental health distress, and secondly, we divide paragraphs to make our literature review’s structure more clear. Moreover, we quoted Godwin’s definition of engineering identity and illustrated the previous findings about engineering identity’s effect on engineering students’ mental health based on Jensen and Cross’s research work in 2018 and 2021 to emphasize the knowledge gap in existing research. And the modification for the discussion section can be found in Point&Response 1.

Point 4: Line 194 “we developed a new instrument”. Are validated? How? Who? The questions of the new instrument are general enough and, in my opinion, it is not correct answer them using a scale, they are question that must be answered with a biographic text and analyse using categories.

Response 4: Our instrument about the degree of tension in interpersonal relationships of female engineering students was developed based on Li et al.(2020) and Darling et al.(2007). In Darling et al.’s research (2007), they measured the quality of interpersonal relationships by friendships, love relationships, relationships with parents, and relationships with other family members. In Li et al.’s research (2020), these were teacher-student relationships and peer relationships. So we choose to measure female engineering students’ interpersonal relationships with self-reported relationship quality with teachers, schoolmates, friends, and intimate friends (eg. love relationships). And before resubmitting the revised version, we made some interviews with female engineering students in our sample population about their status in interpersonal relationships and found the four dimensions mentioned above were important enough.

Point 5: Line 232 What are the different levels?

Response 5: Research within the China educational fields usually divides the universities in China into three categories based on MOE’s supportive policy: 985 institutions are at the top level, 211 institutions are at the secondary level, and others are ordinary institutions. We have illustrated this questions in the revised version.

Referring to the opinions of the reviewer and for a better understanding for international readers, we delete the “levels” and use “types” only to describe the differences between 985, 211, or other ordinary institutions as well as comprehensive or polytechnic universities in our study.

Point 6: Line234 if you have the same questionnaire from male, why did you analyse it? In my opinion it is the weak point of your research. You must compare the female results with the male results of the same questionnaire. I think that your research could be improved if you make this male analysis.

Response 6: Thanks for your advice. We compared the female results with the male results in another related research paper upcoming soon. However, given your kind recommendation, we reported the results briefly within Table3 in the revised manuscript. We found female engineering students tend to have a higher mean score of mental health disorders than men or not-reported gender engineering students (female: 1.46, 1.61, 1.56; male or not-report gender: 1.43, 1.55, 1.51; for mean score of stress, anxiety, and depression respectively). And referring to Jensen and Cross’s work in 2021, we performed Chi-square tests of independence were performed to examine the relationship between gender and the severity levels of stress, anxiety, and depression. The results show there was a significant relationship between gender and stress level (X2(4) = 4.121, p = 0.039, N=1113), the same findings as Foster and Spencer in 2003.

Point 7: Line 300, 301, 302 the results of 13.03, 15.96 and 14.36, how do you calculate? Why are they significant? In the table 4 are different values that I don’t understand.

Response 7: 13.03%=2.66%+3.19%+7.18%, 15.96%=5.05%+1.60%+9.31%, 14.36%=4.52%+0.80%+9.04%, which means the percentage of female engineering students with a moderate, severe and extremely severe level of stress, anxiety, and depression. And as they were descriptive statistics results, so we didn’t report any significance.

Point 8: Line 303, if your values are lower than other studies why are they extremely severe?

Response 8: In table3 (revised version), the severity of self-reported stress, anxiety, and depression (from normal to extremely severe) were calculated by the scores and cutoff values based on the DASS-21 items (Table1), instead of compared with other studies. In our research, a score of 1 to 5 (displayed in Table 1) means normal to extremely severe mental health conditions. The lower scores indicate a lower level of stress, anxiety, or depression, which means a better status of mental health among female engineering students.

Point 9: Line 362 Put together figures 3 and 4 for a better comparison and bigger.

Response 9: Modification done.

Point 10: Line 419 what are the specific results of these studies?

Response 10: Previous studies about the engineering identity concentrated on the concept, structure, dimensions, and measurement of the construct instead of its effect on the mental health of engineering students, which were not very closely related to the current study. As a result, we delete the relevant content in our revised version.

Reviewer 2 Report

General comments;

Overall this is a paper that needs to be published as it adds to the current body of knowledge upon which developments within engineering education are reliant. 

As such it is suggested that some minor additions could be included to assist an international reader understand some aspects of the context without additional work, for example within the introduction the typical age range of 'college' students would help where the same term is used for differing age groups in different nations (the reviewer assumed these students were generally between 18 and 21 years of age?).  Additionally a short footnote giving some brief categorisation of the ST would be appropriate for readers less familiar with the detail.

This paper is sufficiently complete as it stands but an acknowledgement of the potential for the students' degree of neurodiversity to be a contributory factor.  The connection between diagnosed autism spectrum disorders and anxiety has been reported as has the prevelance of ASD within engineers, suggested reading is the work of Prof Simon Baron-Cohen of Cambridge U.

In response to comments within your discussion I would caution against extending into a wider STEM study as the specifics of the subject identities may mask or highlight spurious outcomes that do not deepen understanding, but just generalise, when deepening can support action to reduce negative student experiences.  in a similar way N=376, or 5 institutions, is not specifically a concern as I would see great value in your extending the study beyond the quantitative and into a qualitative study of particular student experiences.  I agree that in many ways engineering has a masculine culture and the status of quantitative studies over qualitative ones may be a reflection of this?  In general if we can use research to indicate who may be the specific students 'at risk' it is qualitative studies that will inform the design of individualised interventions.

Specific comments;

As mentioned above on line 38 a typical age range would help international readers

On line 46 'disciplinary difficulty', is this percieved or promoted? as essentially degree level study in one subject is considered as equal as study in another.  Are you considering that a perception of difficulty is help by the student or a implied level of difficulty is promoted by those within the profession such as may be seen in the medical disciplines? Clearly both may have negative impacts upon student mental health but the corrective actions would differ.

On line 79 you have a name/date citation, is this the same source as given lower as [20]?

Line 92 possibly correct ....by many demographic to and social factors.

Line 201/202 .....suffering from a higher more tense interpersonal relationships.

Overall, thank you for undertaking this study.

Author Response

Dear editor and reviewer:

Thanks very much for taking your time to review this manuscript(ijerph-1851776). We really appreciate all your comments and suggestions! Please find the itemized responses in below and my revisions in the resubmitted files.

Point 1: As such it is suggested that some minor additions could be included to assist an international reader understand some aspects of the context without additional work, for example within the introduction the typical age range of 'college' students would help where the same term is used for differing age groups in different nations (the reviewer assumed these students were generally between 18 and 21 years of age?). Additionally a short footnote giving some brief categorisation of the ST would be appropriate for readers less familiar with the detail.

As mentioned above on line 38 a typical age range would help international readers.

Response 1: Thanks, explanatory information has been added to the revised version. For “college students”, generally, in China undergraduate students were between 18 and 22 years old. In the current study, we recruited both undergraduate students and graduate students, so the average age of college students in this research is 21.86 years, Minimum age =19, Maximum age =28, and SD=1.65. And for “ST”, research within the China educational fields usually divides the universities in China into three categories based on MOE’s supportive policy: 985 institutions are at the top level, 211 institutions are at the secondary level, and others are ordinary institutions. We have illustrated these information in the revised manuscript.

Point 2: This paper is sufficiently complete as it stands but an acknowledgement of the potential for the students' degree of neurodiversity to be a contributory factor.  The connection between diagnosed autism spectrum disorders and anxiety has been reported as has the prevelance of ASD within engineers, suggested reading is the work of Prof Simon Baron-Cohen of Cambridge U.

Response 2: Thanks for your kind advice. We read the research work of Prof. Simon Baron-Cohen (his book Infantile Autism published in 2018 and his open published papers). Autism is a totally new field for us, and we have difficulty fully grasping it in a short time. So we didn’t discuss the effect and association of ASD and anxiety among engineering students in this manuscript. However, we are still curious about such ideas and we add the relevant future research directions in the discussion part.

Point 3: In response to comments within your discussion I would caution against extending into a wider STEM study as the specifics of the subject identities may mask or highlight spurious outcomes that do not deepen understanding, but just generalise, when deepening can support action to reduce negative student experiences.

Response 3: Thanks. We have removed the contents about extending into other STEM disciplines.

Point 4: In a similar way N=376, or 5 institutions, is not specifically a concern as I would see great value in your extending the study beyond the quantitative and into a qualitative study of particular student experiences.  I agree that in many ways engineering has a masculine culture and the status of quantitative studies over qualitative ones may be a reflection of this?  In general if we can use research to indicate who may be the specific students 'at risk' it is qualitative studies that will inform the design of individualised interventions.

Response 4: We really appreciate your professional advice. Given your advice, we add some ideas about the future research design. We would like to make some long-term tracked case studies and deeply observe the whole process of female engineering students’ mental health status and make clearer the mechanism of different identities’ effect on mental health. And we expect further explored how these effects occur and how to negotiate these different identities in order to better deal with mental health crises based on our qualitative research data.

Point 5: On line 46 'disciplinary difficulty', is this percieved or promoted? as essentially degree level study in one subject is considered as equal as study in another. Are you considering that a perception of difficulty is help by the student or a implied level of difficulty is promoted by those within the profession such as may be seen in the medical disciplines? Clearly both may have negative impacts upon student mental health but the corrective actions would differ.

Response 5: We pointed out that ‘disciplinary difficulty’ because “engineering was been seen as one of the toughest and most stressful fields of study” (Danowitz and Beddoes, 2018), just like medical disciplines. And “Feeling overwhelmed due to pace and workload in coursework was a top reason cited by students for leaving STEM programs” (Jensen and Cross, 2018). In our opinion, affected by the course difficulty and employment pressure as well as cut-throat competition, engineering disciplines’ “difficulty is promoted by those within the profession such as may be seen in the medical disciplines”. We would like to help solve female engineering students’ mental health crisis through a gender or professional identity lens, even if we cannot change the ‘disciplinary difficulty’ easily in the short term.

Point 6: On line 79 you have a name/date citation, is this the same source as given lower as [20]?

Response 6: Thank you for pointing out the negligently mistaken citation. We have corrected and updated the citations as well as references.

Point 7:

Line 92 possibly correct ....by many demographic to and social factors.

Line 201/202 .....suffering from a higher more tense interpersonal relationships.

Response 7: Thanks for your recommendations. Corrections have been done.
